# Decoupling Angles and Strength in Low-rank Adaptation

**Massimo Bini**[1,2,3,†]**, Leander Girrbach**[2,3]**, Zeynep Akata**[2,3]
[1]University of Tübingen, Tübingen AI Center,      [2]Helmholtz Munich,
[3]Technical University of Munich, Munich Center for Machine Learning, MDSI
[†]`massimo.bini@uni-tuebingen.de`

## Abstract

Parameter-Efficient FineTuning (PEFT) methods have recently gained significant popularity thanks to the widespread availability of large-scale pretrained models. These methods allow for quick adaptation to downstream tasks with minimal computational cost. However, popular finetuning methods such as LoRA exhibit limited robustness when it comes to hyperparameter choices or extended training regimes, preventing optimal out-of-the-box performance. In contrast, bounded approaches, such as ETHER, provide greater robustness but are limited to extremely low-rank adaptations and fixed-strength transformations, reducing their adaptation expressive power. In this work, we propose Decoupled Low-rank Adaptation (DeLoRA), a novel finetuning method that normalizes and scales learnable low-rank matrices. By bounding the distance of the transformation, DeLoRA effectively decouples the angular learning from the adaptation strength, enhancing robustness without compromising performance. Through evaluations on subject-driven image generation, natural language understanding, and instruction tuning, we show that DeLoRA matches or surpasses performance of competing PEFT methods, while exhibiting stronger robustness. Code is available at `https://github.com/ExplainableML/DeLoRA`.

## 1 Introduction

The rapid advancement of deep learning has led to the development of large-scale pretrained models in various domains, especially in computer vision and natural language processing (Touvron et al., 2023a;b; Radford et al., 2021; Rombach et al., 2022). However, the enormous size of these models, reaching billions of parameters, presents significant challenges when adapting them to specific downstream tasks, particularly in terms of computational cost and efficiency. To address these challenges, Parameter Efficient FineTuning (PEFT) methods have emerged. PEFT methods are characterized by their introduction of a small set of learnable parameters, in contrast to the extensive parameter updates required in full finetuning. Notable examples include adapters (Houlsby et al., 2019) and prompt tuning (Lester et al., 2021). In this work, we focus on enhancing LoRA (Hu et al., 2022), a widely adopted finetuning method known for its simplicity and effectiveness. However, despite its success, LoRA is sensitive to hyperparameter choices (Biderman et al., 2024) and often exhibits performance degradation during extended finetuning (Qiu et al., 2023). While robust finetuning approaches such as ETHER and ETHER+ (Bini et al., 2024) address some of these limitations, they are constrained to extremely low-rank adaptations and fixed-strength transformations.

Therefore, we propose DeLoRA, an enhanced version of LoRA that introduces a boundary on the weight updates through normalization, decoupling the angular learning from the adaptation strength. This enhances adaptability across diverse settings while preserving capabilities for personalization and merging at inference time. We motivate DeLoRA from two distinct perspectives: as an extension of LoRA through the introduction of additional normalization, and as an evolution of ETHER by enabling high-rank updates. We conduct ablation studies on the design choices and demonstrate improvements over both LoRA and ETHER. Additionally, we validate the advantages of DeLoRA by evaluating it across diverse tasks in image generation and LLM adaptation.

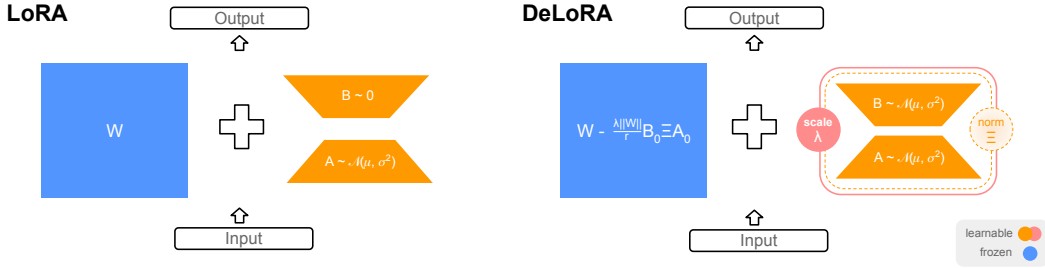

Figure 1: Visualizations (Left) of the original LoRA (Hu et al., 2022) and (Right) of our proposed method DeLoRA. In addition to the low-rank matrices $B, A$, we introduce a normalization $\Xi$ and a scaling factor $\lambda$, which effectively decouple the angular learning from the adaptation strength.

In summary, we make the following contributions in this work: (i) we thoroughly review the formulations of LoRA and ETHER and derive a novel PEFT method, DeLoRA; (ii) we demonstrate DeLoRA enhanced robustness and decoupling compared to alternatives; (iii) we extensively ablate the formulation of DeLoRA by deriving it from both LoRA and ETHER; (iv) we evaluate DeLoRA on both vision and language benchmarks, matching or surpassing the performance of competing PEFT methods.

## 2 DECOUPLED LOW-RANK ADAPTATION (DELORA)

Our decoupled low-rank adaptation approach, by introducing learnable boundaries on the weight updates, effectively combines the strengths of LoRA and ETHER methods, allowing for high expressivity and finetuning robustness. In the following sections, we will (i) present an overview of the PEFT methods LoRA and ETHER, focusing on their respective limitations (Section 2.1) and (ii) describe how we derive our proposed DeLoRA method from both perspectives (Section 2.2), along with a comparison with DoRA (Liu et al., 2024a), a method that also targets decoupling angular and magnitude components.

### 2.1 PRELIMINARIES: LORA & ETHER, AND THEIR LIMITATIONS

Here, we provide a detailed review of LoRA (Hu et al., 2022) and ETHER (Bini et al., 2024), with a particular focus on their limitations.

**Low-rank Adaptation (LoRA).** Hu et al. (2022) proposed *Low-rank Adaptation (LoRA)*, which parametrizes the update of pretrained weights $W \in \mathbb{R}^{d \times f}$ during finetuning as

$$\left(W + \frac{\alpha}{r} BA\right)^\mathsf{T} x + b \tag{1}$$

where $A \in \mathbb{R}^{r \times d}$ and $B \in \mathbb{R}^{f \times r}$ are the learnable matrices, $\alpha$ is a scaling factor, and $r$ is the rank of the final $BA$ matrix. When $r \ll \min(d, f)$, LoRA substantially reduces the number of parameters required for finetuning compared to full finetuning. Furthermore, $BA$ matrices can be integrated into $W$ at inference time, eliminating additional latency.

However, LoRA is known to be highly sensitive to hyperparameter choices (Biderman et al., 2024), and it is prone to deterioration with over-training (Qiu et al., 2023), thus requiring careful tuning and experimentation to achieve an optimal balance between a sufficiently high learning rate and avoiding catastrophic overwriting of the pretrained weights. In our proposed DeLoRA, we mitigate this behavior by introducing a boundary to the weight updates, which results in robust performance across a broad range of learning rates.

**Finetuning with Hyperplane Reflections (ETHER).** Following efficiency and robustness arguments, Bini et al. (2024) propose to employ bounded transformations for finetuning, namely ETHER

and ETHER+. ETHER (left side in Eq. (2)) and ETHER+ (right side) introduce multiplicative transformations $H$ or $H^+$ respectively, which act on the pretrained weights as follows:

$$(HW)^\intercal x + b \quad , \quad \left(H^+ W \tilde{H}^+\right)^\intercal x + b. \tag{2}$$

Here, $H = I - 2uu^\intercal$, $H^+ = I - uu^\intercal + vv^\intercal$, $\tilde{H}^+ = I - \tilde{u}\tilde{u}^\intercal + \tilde{v}\tilde{v}^\intercal$ (where $u$, $v$, $\tilde{u}$, $\tilde{v}$ are unit vectors) are bounded in terms of their distance to the identity transformation, as per

$$\|H - I\|_F = 2 \quad , \quad \|H^+ - I\|_F \leq 2, \tag{3}$$

where the subscript $_F$ denotes the Frobenius norm. This upper bound on the transformation distance prevents weight changes that cause catastrophic overwriting, as shown by Bini et al. (2024).

However, enforcing a constant boundary on the transformation distance can limit the finetuning performance, as the boundary may be too strict to adapt the layer or pretrained model at hand to the respective task. Furthermore, by rewriting the formulations in Eq. (2) in a residual form, we can show that the weight updates are intrinsically limited to be low-rank (see Appendix A), which limits the finetuning capacity of such methods. In DeLoRA, by introducing a normalization and a scaling factor to LoRA matrices, we show how to achieve robustness comparable to ETHER while enabling control over both boundary and rank, ultimately enhancing model expressivity and performance.

## 2.2 DeLoRA

While both LoRA and ETHER demonstrate valuable properties, namely parameter efficiency and robustness, they also exhibit notable limitations. Our proposed PEFT method, DeLoRA, addresses these shortcomings by synthesizing the strengths of both approaches. In this regard, DeLoRA can be thought of as an extension of LoRA that incorporates ETHER's robustness properties or, alternatively, as an enhancement of ETHER that adopts LoRA's more expressive paradigm. In the following, we will present both derivations and finally summarize in a concise way our proposed DeLoRA formulation.

**Deriving DeLoRA from LoRA.** In order to achieve robustness to learning rates, we first observe that in LoRA's Eq. (1) the norm of the weight updates $\Delta W$ is proportional to $\Delta BA$, which in turn is proportional to the learning rate. This means that the update strength at each training step is directly driven by the learning rate, which can lead to catastrophic overwriting in high learning rate regimes. In order to mitigate this behavior, we want to introduce a normalization term. To do this, we start by decomposing the $BA$ matrix into the sum of its rank-1 components, i.e.

$$BA = \sum_{i=1}^{r} b_i a_i^\intercal \tag{4}$$

■ *Controllable Boundary.* Similarly to ETHER, we normalize each rank-1 entry, making the Frobenius norm of each single rank-1 component equal to 1. This normalization can be introduced as in

$$\sum_{i=1}^{r} \frac{b_i a_i^\intercal}{\|b_i\|\|a_i\|} = B\Xi A \tag{5}$$

where $\Xi$ is a diagonal matrix with entries $\Xi_{i,i} = \frac{1}{\|b_i\|\|a_i\|}$ for $i = 1, \ldots, r$, $\Xi_{i,j} = 0$ for $i, j = 1, \ldots, r, i \neq j$. The final update distance with respect to the pretrained weights thus is bounded as

$$\|B\Xi A\| = \left\|\sum_{i=1}^{r} b_i a_i^\intercal\right\| \leq \sum_{i=1}^{r} \|b_i a_i^\intercal\| = r \tag{6}$$

Most importantly, the boundary is independent of the used learning rate. Next, to control the boundary and remove its rank dependency, we scale $B\Xi A$ by a factor $\frac{\lambda}{r}$, as in

$$\left\|\frac{\lambda}{r} B\Xi A\right\| \leq \lambda. \tag{7}$$

Now, the boundary is equal to $\lambda$ and can be chosen arbitrarily to better fit the pretrained network or task at hand. To enable greater flexibility and layer-specific boundaries, we make each distinct $\lambda$

learnable, allowing finetuning to adapt their values accordingly. Hence, we effectively decouple the angular learning (the normalized $B\Xi A$ matrices) from the adaptation strength, as measured by the boundary $\lambda$. Furthermore, introducing a single additional learnable parameter $\lambda$ to each finetuned matrix creates only negligible overhead in terms of overall trainable parameters and training speed.

■ *Weights-norm Scaling.* Previous works suggest that when finetuning image generative models such as Stable Diffusion, multiplicative finetuning methods exhibit stronger performance (Qiu et al., 2023; Liu et al., 2024b) than additive counterparts. We argue this may arise because multiplicative methods induce weight updates relative to the pretrained weights $W$, meaning updates are inherently layer-specific. This might be especially relevant when adapting a diverse set of layers, which is the case for our Stable Diffusion adaptations (see Fig. 3). To mimic this approach, in our additive proposed method DeLoRA, we introduce a scaling factor equal to the pretrained weights norm. This can be formally stated as

$$\Delta W = \frac{\lambda \|W\|}{r} B\Xi A. \tag{8}$$

Our ablation studies on Stable Diffusion finetuning tasks demonstrate such performance improvements empirically (see Section 3.2).

■ *Initialization.* To initialize the finetuning process from the pretrained model, DeLoRA's normalization operation does not allow to simply follow LoRA's zero initialization of the $B$ matrix. From preliminary experiments, we find that introducing a small epsilon to avoid division by 0, would sometimes lead to unstable results. Therefore, we instead follow (Meng et al., 2024; Bini et al., 2024) and subtract a copy of the kaiming-randomly initialized matrices to the frozen pretrained weights, as in

$$W = \bar{W} - \left( \frac{\lambda \|W\|}{r} B\Xi A \right)_0 \tag{9}$$

where $\bar{W}$ is the original pretrained matrix, and $(\frac{\lambda \|W\|}{r} B\Xi A)_0$ is the update matrix at time 0.

**Deriving DeLoRA from ETHER**    So far, we showed how to derive DeLORA from LoRA. Alternatively, it is possible to derive DeLoRA by introducing properties of LoRA to ETHER. We find this to be insightful to understand the impact of each individual component from a theoretical perspective. In addition, we quantitatively ablate all innovations of DeLoRA in Section 3.2.

■ *Controllable Boundary.* One of the primary limitations of ETHER and ETHER+ is their fixed boundary (see Section 2.1), which is fixed and thus cannot be adapted to the pretrained model in use. We address this limitation by introducing a scaling parameter $\lambda$ as in

$$H = I - \lambda uu^\mathsf{T} \quad , \quad H^+ = I - \frac{\lambda}{2} uu^\mathsf{T} + \frac{\lambda}{2} vv^\mathsf{T}. \tag{10}$$

Consequently, the boundaries on the distances of $H$ and $H^+$ from the identity matrix become $\|H - I\|_F = \lambda$, and $\|H^+ - I\|_F \leq \lambda$. In Section 3.2, we show that this modification, i.e. introducing a controllable bound, leads to the largest increase in performance.

■ *Increasing the rank.* In Appendix A, we demonstrate that ETHER and ETHER+ are restricted to rank-1 and rank-4 weight updates respectively. In order to arbitrarily control the rank, we extend the $H^+$ parameter of ETHER+ to $\hat{H}$, which allows for an arbitrary number of weight reflection operations:

$$\hat{H} = I - \sum_{i=1}^{r/2} u_i u_i^\mathsf{T} + \sum_{i=1}^{r/2} v_i v_i^\mathsf{T}. \tag{11}$$

We can rewrite $\hat{H}$ by gathering the $u$ and $v$ unit vectors into two rank-$\frac{r}{2}$ matrices, as in

$$\hat{H} = I - U\Sigma U^\mathsf{T} + V\Theta V^\mathsf{T}, \tag{12}$$

where $\Sigma$ and $\Theta$ are diagonal normalization matrices with entries $\Sigma_{i,i} = \frac{1}{\|u_i\|^2}$, $\Theta_{i,i} = \frac{1}{\|v_i\|^2}$, The entries on the diagonals of $\Sigma$ and $\Theta$ are constructed to normalize $u$ and $v$ to unit vectors. Thus, the distance from the identity matrix becomes

$$\|\hat{H} - I\| = \left\| \sum_{i=1}^{r/2} u_i u_i^\mathsf{T} - \sum_{i=1}^{r/2} v_i v_i^\mathsf{T} \right\| \leq \sum_{i=1}^{r/2} \|u_i u_i^\mathsf{T}\| + \sum_{i=1}^{r/2} \|v_i v_i^\mathsf{T}\| = r. \tag{13}$$

As above, we can control the boundary on the distance, and remove the rank dependency, by introducing a scaling factor $\frac{\lambda}{r}$ as in

$$\hat{H} = I - \frac{\lambda}{r}U\Sigma U^{\intercal} + \frac{\lambda}{r}V\Theta V^{\intercal} \tag{14}$$

■ *U,V Relaxation.* Finally, we relax $U\Sigma U^{\intercal}, V\Theta V^{\intercal}$ and replace them with distinct trainable matrices $B\Xi A$ and $D\Phi C$ respectively, which leads to $\hat{H} = I - \frac{\lambda}{r}(B\Xi A - D\Phi C)W$. We emphasize how this formulation resembles a multiplicative analog of our proposed DeLoRA method, and we include this variant in our ablation study. We ablate all alternatives in Section 3.2. There, we find that DeLoRA, combined with weights-norm scaled updates, as in multiplicative finetuning, achieves overall stronger performance.

**DeLoRA formulation.** Summarizing, our proposed DeLoRA finetuning method consists in learning a normalized low-rank matrix $B\Xi A$ and a scale $\lambda$, updating the pretrained weights as in

$$\left(W + \frac{\lambda\|\bar{W}\|}{r}B\Xi A\right)^{\intercal} x + b \tag{15}$$

This formulation inherently constrains the learnable finetuning updates in a $\lambda\|\bar{W}\|$-sized ball, where $\bar{W}$ is the norm of the pretrained weights, achieving a decoupling of the transformation strength from the angular learning.

In more detail, the key components are:

- *Normalization:* $\Xi$ is a $r$-dimensional diagonal matrix that normalizes LoRA's inner low-dimensional bottleneck (Eq. (5)), bounding the Frobenius norm of $B\Xi A$ to $r$ (Eq. (6)).
- *Scaling Factors:* (i) $1/r$ is used to remove the rank dependency on the boundary dimensionality, (ii) $\|\bar{W}\|$ to make the weight updates proportional to the pretrained weights, and (iii) $\lambda$ to control the adaptation strength and allow for a layer-specific boundary adaptation (Eq. (7))
- *Initialization:* Pretrained initialization follows by merging to the pretrained weights a frozen copy of the initialized finetuning adaptation matrices (Eq. (9)).

**DoRA vs DeLoRA discussion.** DoRA (Liu et al., 2024a), similarly to our work, addresses finetuning targeting the decoupling of angular and magnitude components, by using a formulation that leads to weight updates $W' = m\frac{W+\Delta W}{\|W+\Delta W\|}$. We can summarize the key differences between DoRA and our proposed method in two main aspects: (i) DoRA applies normalization and scaling operations on the fully finetuned weights, and (ii) these operations are performed on the column space of the weight matrices, which significantly differs from our approach. In contrast, we argue that DeLoRA finetuning has two key advantages: (i) by introducing the normalization and scaling operations directly on the weight updates $\Delta W$, it more effectively prevents divergence from the pretrained model, and (ii) by normalizing the inner low-dimensional space (as opposed to the column space), it implicitly enforces a Frobenius-norm boundary, providing a mathematical guarantee against divergence. These ultimately result in (i) peculiar training dynamics (as depticted in Fig. 3, whereas DoRA and LoRA exhibit similar behavior), and (ii) enhanced decoupling, supported by the robustness performance in Fig. 2 and in Appendix C. In this regard, we notice that although DeLoRA's learnable boundary theoretically allows an unbounded Frobenius norm, divergence from the pretrained weights does not happen in practice, as also shown in Appendix D. This demonstrates that during finetuning, DeLoRA's learnable boundary is able to effectively adjust and avoid divergence from the pretrained weights–behavior that is not observed with DoRA.

## 3 EXPERIMENTS

In this section, we evaluate our proposed DeLoRA method for image generation, natural language understanding, and instruction tuning tasks. We begin by providing a detailed description of these tasks and their relevance. To justify our design choices, we present a comprehensive ablation study that highlights the key innovations of DeLoRA. Finally, we demonstrate that DeLoRA not only matches or exceeds the performance of LoRA and other state-of-the-art methods but also exhibits superior robustness. This enhanced stability is particularly evident in two aspects: reduced sensitivity to learning rate selection and improved performance retention during extended finetuning periods.

## 3.1 TASKS

**Subject-driven Image Generation.** Following (Qiu et al., 2023; Bini et al., 2024), we assess the effectiveness of our proposed methods in the DreamBooth setting (Ruiz et al., 2023), specifically by adapting Stable Diffusion (Rombach et al., 2022) to recontextualize a subject shown in a set of images according to a given prompt. The dataset, sourced from (Ruiz et al., 2023), comprises 30 subjects, each paired with 25 prompts. The task is to finetune Stable Diffusion to generate images portraying the given subject in the context defined by the prompts. We report an example in in Appendix E (Fig. 7, left side). For each combination of image and prompt, after finetuning, we generate four images and measure the subject-fidelity by DINO (Caron et al., 2021) and CLIP (Radford et al., 2021), as proposed by (Ruiz et al., 2023). Here, the score represents the similarity of generated and given images, measuring the faithfulness of generating images of the given subject to the provided real images. Among the two metrics, the DINO score is more significant since it is more sensitive to subject-unique features (Ruiz et al., 2023).

**Semantic Map to Image** Following (Qiu et al., 2023; Bini et al., 2024), we evaluate the ability of our proposed methods in finetuning Stable Diffusion to generate realistic images based on given segmentation maps. The image should follow the spatial structure laid out in the segmentation map as closely as possible. Examples of segmentation maps and their corresponding generated images are presented in Appendix E (Fig. 7, right side). For the control signal, we use the pretrained encoder from ControlNet (Zhang et al., 2023a). For training and evaluation, we utilize semantic maps and images from the ADE20K dataset (Zhou et al., 2019). After training, we generate images for 2000 segmentation masks from the ADE20K validation set and report the mean Intersection-over-Union (mIoU) and accuracy of semantic maps as predicted by UperNet-101 (Xiao et al., 2018). Note that we only use the Semantic Map to Image task to ablate our method design decisions.

**Natural Language Understanding** We evaluate DeLoRA's performance in adapting small-scale language models by finetuning and evaluating a pretrained RoBERTa-base model (Liu et al., 2020) on the General Language Understanding Evaluation (GLUE) benchmark (Wang et al., 2018). GLUE tasks have been extensively used to measure natural language understanding performance, comprising inference tasks (MNLI, QNLI, RTE), sentiment classification (SST-2), and correct identification of English grammatical structures (CoLA). CoLA results refer to Matthews correlation coefficient, MNLI to matched accuracy, and STS-B to average correlation, while all other tasks are evaluated on accuracy. For a proper evaluation on the validation set, we adopt the setup proposed by Wu et al. (2024b), and split the validation set into two subsets, guarded by a pre-defined seed, that will be used for model selection and evaluation. We provide more details in Section 3.3.

**Instruction Tuning.** We evaluate how effectively DeLoRA can adapt LLMs to follow user-given instructions, finetuning LLaMA-2-7B (Touvron et al., 2023b) on the Alpaca dataset (Taori et al., 2023). Following Bini et al. 2024, we evaluate the zero-shot performance of instruction-tuned models on four different tasks, namely (1) Massive Multitask Language Understanding (MMLU) (Hendrycks et al., 2021), which features 57 tasks in different categories such as STEM, Humanities, and Social Sciences; (2) AI2 Reasoning Challenge (ARC) (Clark et al., 2018), which contains over 7000 grade-school science questions; (3) TruthfulQA (Lin et al., 2022), which contains 817 questions representing common misconceptions in 38 categories like health, law, finance and politics. TruthfulQA additionally features two separate sub-tasks, namely single-true and multi-true. In single-true, only one of the provided answers is correct, and the model has to select the unique correct answer. In multi-true, several of the provided answers may be correct, and the model has to assign a high probability to correct answers and a low probability to incorrect answers.

## 3.2 ABLATION OF DELORA DESIGN CHOICES

In this section, we ablate the incremental design choices that transform LoRA and ETHER+ into DeLoRA, evaluating these on the subject-driven generation and semantic map-to-image tasks. From the LoRA derivation (top-down in Tables 1,2), we show how incorporating normalization with a controllable boundary and weight scaling into pretrained matrices enhances performance. From the ETHER+ derivation (bottom-up in Tables 1,2), we show how introducing a controllable scale, a higher-rank formulation, relaxed learnable matrices, and an additive finetuning tranformation, incrementally improves performance.

| Method | | $\Delta W$ formulation | DINO | CLIP-I |
|---|---|---|---|---|
| **LoRA** [rank-$r$] | | $BA$ | 0.674 | 0.785 |
| ↓ + normalize w/ controllable boundary | | $\frac{\lambda}{r}B\Xi A$ | 0.682 | 0.809 |
| · + normalize w/ controllable boundary + weights-scaling 
 · + controllable boundary + high rank + relaxed + additive FT | **(DeLoRA)** | $\frac{\|W\|\lambda}{r}B\Xi A$ | **0.701** | 0.825 |
| ↑ + controllable scale + high rank + relaxed | | $\frac{\lambda}{r}(B\Xi A - D\Phi C)W$ | 0.696 | 0.833 |
| \| + controllable boundary + high rank | | $\frac{\lambda}{r}(U\Sigma U^\intercal - V\Theta V^\intercal)W$ | 0.685 | **0.840** |
| \| + controllable boundary | | $\lambda(uu^\intercal - vv^\intercal)W$ | 0.678 | 0.810 |
| **ETHER+ (one-sided)** [rank-2, boundary equal to 2] | | $(uu^\intercal - vv^\intercal)W$ | 0.624 | 0.746 |

Table 1: Ablation of DeLoRA innovations on the **Subject-driven Image Generation** task. We show how different components affect performance from both LoRA and ETHER derivation.

| Method | | $\Delta W$ Formulation | mIoU ↑ | Acc. ↑ | FID ↓ |
|---|---|---|---|---|---|
| **LoRA** [rank-$r$] | | $BA$ | 25.13 | 64.95 | 31.35 |
| ↓ + normalize w/ controllable boundary | | $\frac{\lambda}{r}B\Xi A$ | 25.66 | **65.82** | 31.01 |
| · + normalize w/ controllable boundary + weights-scaling 
 · + controllable boundary + high rank + relaxed + additive FT | **(DeLoRA)** | $\frac{\|W\|\lambda}{r}B\Xi A$ | **26.10** | 65.08 | 30.71 |
| ↑ + controllable boundary + high rank + relaxed | | $\frac{\lambda}{r}(B\Xi A - D\Phi C)W$ | 25.55 | 65.16 | **29.89** |
| \| + controllable boundary | | $\lambda(uu^\intercal - vv^\intercal)W$ | 24.56 | 62.70 | 31.28 |
| **ETHER+ (one-sided)** [rank-2, boundary equal to 2] | | $(uu^\intercal - vv^\intercal)W$ | 23.46 | 62.26 | 31.18 |

Table 2: Ablation of DeLoRA innovations on the **Semantic Map to Image** task. We show how different components from both LoRA and ETHER derivations incrementally improve performance.

Results for subject-driven image generation are in Table 1. For this ablation we use a small-scale version of the setting proposed by (Ruiz et al., 2023), finetuning 3 subjects over 25 prompts each (10% of the data). Among all modifications, we notice how the introduction of a controllable boundary in ETHER+ (one-sided) has the highest impact, raising the DINO score from 0.624 to 0.678 and the CLIP score from 0.746 to 0.810. This shows how the lack of strength is the hindering factor for ETHER+(one-sided), as already noted by (Bini et al., 2024). Starting from LoRA, we notice how the weights-norm scaling has the largest impact on performance, raising the DINO score from 0.682 to 0.701 and the CLIP score from 0.809 to 0.825. Additionally, we note that DeLoRA's performance without the weights-norm scaling falls short compared to its multiplicative counterpart.

For the Semantic Map to Image ablation study, we run a small-scale grid search by finetuning Stable Diffusion for 10 epochs on ADE20K in bfloat16 precision. Results are reported in Table Table 2. We note how DeLoRA achieves best controllability among different variations. In addition, we also note the increase in Accuracy when increasing the rank of ETHER+, hinting that it could have been a limiting factor.

## 3.3 BENCHMARK RESULTS

**Subject-Driven Image Generation** Results are in Table 3. For a comprehensive benchmark performance comparison, we report low-rank results from Bini et al. (2024), while running and evaluating LoRA, DoRA, and DeLoRA methods at a consistent rank. For each method, we conduct a grid search to identify optimal hyperparameters using the same 3 subjects as in the ablation studies, then evaluate the top-performing configurations on the full 30-subject benchmark, testing each across three distinct seeds. The best and average results are reported in Table 3. We notice that LoRA, DoRA, and DeLoRA, all achieve comparable average performance in terms of DINO and CLIP-Image, all outperforming lower-rank baselines. This shows that DeLoRA is able to effectively combine ETHER+ robustness properties with superior performance.

**Natural Language Understanding** Results are in Table 4. For proper evaluation on the GLUE validation set, we follow Wu et al. (2024a;c) and split the validation set into two subsets (determined by pre-defined seeds), and use the first subset to tune hyperparameters, and the second subset to evaluate method performance. For fair comparisons we use same seeds as Wu et al. (2024a;c). In addition, in order to compare with LoRA's implementation, we simply apply DeLoRA to Q,V attention layers with rank 8, which is likely sub-optimal with respect to applying lower-rank modules

| Method | | #param | DINO | CLIP-I |
|---|---|---|---|---|
| Real Images | | | 0.703 | 0.864 |
| DreamBooth | (Ruiz et al., 2023) | 859.5M | 0.644 | 0.793 |
| $OFT_{n=4}$ | (Qiu et al., 2023) | 11.6M | 0.652 | 0.794 |
| ETHER+ | (Bini et al., 2024) | 0.4M | 0.666 | 0.800 |
| $LoRA_{r=4}$ | (Hu et al., 2022) | 0.8M | 0.660 | 0.796 |
| $LoRA_{r=16}$ | (Hu et al., 2022) | 3.2M | 0.686 | 0.818 |
| $DoRA_{r=16}$ | (Liu et al., 2024a) | 3.2M | **0.687** | 0.819 |
| $DeLoRA_{r=16}$ | (ours) | 3.2M | 0.686 | **0.820** |
| $LoRA^{\dagger}_{r=16}$ | (Hu et al., 2022) | 3.2M | 0.688 | 0.818 |
| $DoRA^{\dagger}_{r=16}$ | (Liu et al., 2024a) | 3.2M | 0.689 | 0.819 |
| $DeLoRA^{\dagger}_{r=16}$ | (ours) | 3.2M | **0.693** | **0.820** |

Table 3: Results for evaluating DeLoRA in **subject-driven image generation**. † indicates experiments with tuned hyperparameters.

| Method | | #param | MNLI | SST-2 | MRPC | CoLA | QNLI | QQP | RTE | STS-B | Avg |
|---|---|---|---|---|---|---|---|---|---|---|---|
| Full Finet. | | 125M | 87.3 | 94.4 | 87.9 | 62.4 | 92.5 | 91.7 | 78.3 | 90.6 | 85.6 |
| BitFit | (Zaken et al., 2022) | 0.1M | 84.7 | **94.0** | 88.1 | 54.0 | 91.0 | 87.3 | 69.8 | 89.5 | 82.3 |
| IA3 | (Liu et al., 2022) | 0.06M | 85.4 | 93.4 | 86.4 | 57.8 | 91.1 | 88.5 | 73.5 | 88.5 | 83.1 |
| LoReFT | (Wu et al., 2024c) | 0.02M | 83.1 | 93.4 | **89.2** | 60.4 | 91.2 | 87.4 | **79.0** | 90.0 | 84.2 |
| RED | (Wu et al., 2024a) | 0.02M | 83.9 | 93.9 | **89.2** | 61.0 | 90.7 | 87.2 | 78.0 | 90.4 | 84.3 |
| LoRA | (Hu et al., 2022) | 0.3M | 86.6 | 93.9 | 88.7 | 59.7 | **92.6** | 90.4 | 75.3 | 90.3 | 84.7 |
| Adapter^FFN | (Pfeiffer et al., 2021) | 0.3M | **87.1** | 93.0 | 88.8 | 58.5 | 92.0 | 90.2 | 77.7 | 90.4 | 84.7 |
| Adapter | (Houlsby et al., 2019) | 0.4M | 87.0 | 93.3 | 88.4 | 60.9 | 92.5 | **90.5** | 76.5 | 90.5 | 85.0 |
| DeLoRA | (ours) | 0.3M | 86.9 | 93.7 | 88.6 | **64.7** | **92.6** | 90.2 | 77.3 | **90.6** | **85.6** |

Table 4: Comparisons of different methods finetuning RoBERTa-base on **GLUE benchmark**. Results of all baselines are taken from Wu et al. (2024a) and Wu et al. (2024c).

to a larger set of layers (Hu et al., 2022). We notice how DeLoRA achieves better performance on CoLA, QNLI and STS-B, and an overall significantly better average score with respect to all baselines, demonstrating its efficacy in adapting language models for NLU tasks.

**Instruction Tuning** Results are in Table 5. Results for all methods but DoRA and DeLoRA are reported from Bini et al. (2024). For these two, a proper grid search has been run following the same setup of Bini et al. (2024). Further details cab be found in B. We can see that DeLoRA achieves best results on three out of four tasks. This confirms the effectiveness of our improvements, which lead to optimal average performance in this setup. On the MMLU task, ETHER and ETHER+ outperform other methods, but fall short on other tasks, achieving lower average performance compared to DeLoRA. This might be due to the limited capacity of ETHER methods from their rank limitation.

## 3.4 INSIGHTS

In this section we analyze (i) the learning rate robustness properties, and (ii) the training dynamics, with a focus on prolonged training setting, of DeLoRA with respect to other finetuning methods. Then, we analyze (iii) how weights norms differ in a pretrained model, to better understand the weights-norm scaling effect in DeLoRA.

**Learning Rate Robustness.** We conducted a comprehensive learning rate robustness analysis in the setting of the Subject-driven Generation task of Section 3. Evaluation is done reporting DINO scores (Fig.2, Left) and Euclidean distance between finetuned and pretrained weights of a projection layer in an attention module (Fig.2, Right) across multiple methods, using a range of learning rates derived from each method's base learning rate. Our analysis shows that DeLoRA is able to achieve the same robustness of ETHER+, while improving performance, whereas both LoRA and DoRA performance degrade at $4\times$ the base learning rate. We also notice how LoRA updates' distance grows at higher learning rates, while interestingly DoRA, after $8\times$, does not diverge further, likely thanks to its magnitude control. However this does not lead to better performance in these regimes.

| Method | | #param | MMLU | ARC | Tru-1 | Tru-2 | Avg |
|---|---|---|---|---|---|---|---|
| LLaMA-2-7B | | - | 41.81 | 42.92 | 25.21 | 38.95 | 37.22 |
| $ETHER_{n=32}$ | (Bini et al., 2024) | 0.26M | 44.57 | 45.14 | 27.91 | 41.83 | 39.86 |
| $ETHER+_{n=32}$ | (Bini et al., 2024) | 1.04M | **44.87** | 46.50 | 29.38 | 43.51 | 41.07 |
| $LoRA_{r=8}$ | (Hu et al., 2022) | 4.19M | 43.61 | 46.16 | 28.76 | 42.21 | 40.19 |
| $DoRA_{r=8}$ | (Liu et al., 2024a) | 4.19M | 43.24 | 47.18 | 29.01 | 43.47 | 40.73 |
| $DeLoRA_{r=8}$ | (ours) | 4.19M | 44.21 | **47.70** | **29.62** | **44.14** | **41.42** |

Table 5: Results for **Instruction Tuning** on MMLU, ARC, and TruthfulQA benchmarks. Values represent accuracy scores achieved by different finetuning methods. Best scores are highlighted in bold, and second-best scores are underlined.

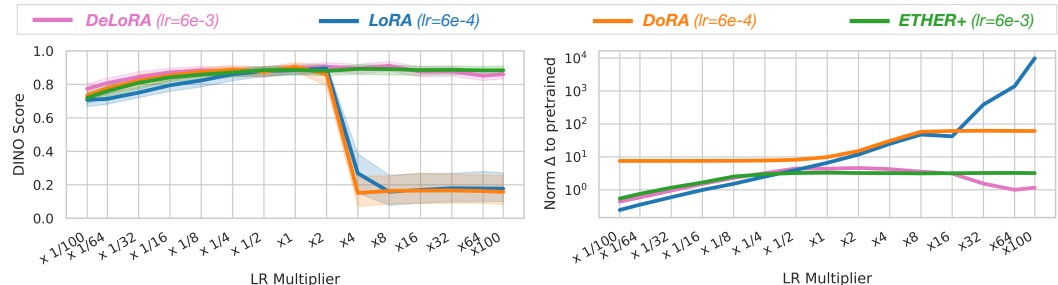

Figure 2: Learning rate robustness plots in Subject-driven generation task in terms of DINO scores (Left) and Euclidean distance between a finetuned vs pretrained projection layer weights (Right). Learning rates used for robustness evaluation were derived by multiplying the base learning rate in a range of factors.

**Finetuning Regime and Prolonged Training.** We further investigate the behavior of weight updates across different methods by measuring the Euclidean distance between finetuned weight matrices (after merging) and the pretrained corresponding matrices during fine-tuning. This provides us a quantitative measure of the shift and rate at which fine-tuned weight matrices diverge from the pretrained weights. In Fig. 3 (Left), we show this analysis for the out-projection matrix in one of StableDiffusion's Unet self-attention layers. We find that LoRA- and DoRA-trained weights continuously depart from the pretrained weights over the course of training, passing through an optimal regime but eventually overshooting and ending in a diverging regime (notice that best performance are typically found between 1000 and 1400 steps). In contrast, DeLoRA-trained weights exhibit a peculiar behavior, quickly moving away from the pretrained weights, until they reach the boundary, from which they cannot diverge further. We argue that this leads to prolonged training robustness, effectively avoiding catastrophic overwriting. Qualitative examples are provided in Fig. 3 (Right) and in Appendix E. Additionally, we highlight that by adjusting the boundary parameter $\lambda$, one can easily control the maximum allowable shift and, therefore, the level of finetuning robustness.

**Weights Norms Heterogeneity.** In Fig. 4, we show the mean of column norms for weight matrices in different attention blocks of the U-Net in Stable Diffusion v1.5. By doing so, we highlight the effect of weights-norm scaling as introduced in Section 2. We find that different modules, as well as different positions in the U-Net, show systematic differences with respect to weights norms. This points at differences within the pretrained model which finetuning methods should account for. Our proposed scaling is one possibility to accomplish this. Exploring more sophisticated methods to include layer-wise differences is an interesting direction for future research.

## 4  RELATED WORK

Parameter efficient finetuning (PEFT) is an active field of research, encompassing methods such as adapters (Houlsby et al., 2019), prompt- and prefix-tuning variations (Lester et al., 2021; Li & Liang, 2021; Liu et al., 2023), and more specialized methods such as BitFit (Zaken et al., 2022), FourierFT (Gao et al., 2024), and LayerNorm Tuning (Zhao et al., 2024). In this paper, we propose an improved

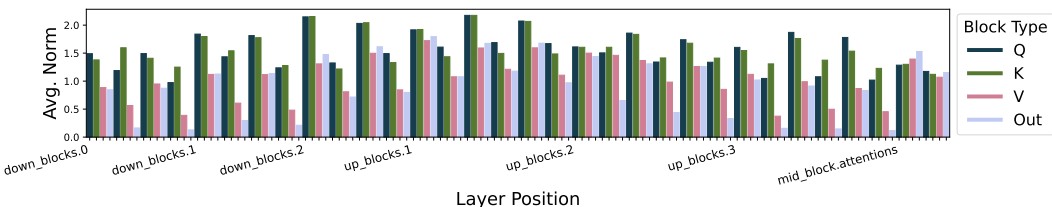

Figure 3: (Left) Euclidean Distance of finetuned weights to pretrained weights as a function of the number of training steps. (Right) Qualitative examples show that LoRA exhibits significant artifacts earlier in the process compared to DeLoRA, which maintains better image quality.

Figure 4: Average column norms of parameters in the attention modules of Stable Diffusion's Unet

PEFT method based on low-rank adapters (LoRA) first described by (Hu et al., 2022). Therefore, we focus our review of previous work on LoRA variants and refer to recent surveys (Han et al., 2024; Xin et al., 2024) regarding PEFT methods in general. LoRA is a popular finetuning approach for large models, featuring advantages such as low-memory footprint and no additional inference cost (Hu et al., 2022). Compared to full-finetuning, LoRA is also less prone to catastrophic forgetting (Biderman et al., 2024).

However, beyond falling behind in performance on downstream tasks compared to full finetuning (Biderman et al., 2024), previous work has identified and attempted to address different limitations of the original LoRA method. Lialin et al. (2023); Zi et al. (2023); Xia et al. (2024); Ren et al. (2024) propose methods to overcome the low-rank limitation without sacrificing memory efficiency. Similarly, VeRA (Kopiczko et al., 2024) keeps the original LoRA setup but reduces trainable parameters further by only scaling the randomly initialized matrices, which are shared across layers. To account for differences between layers, (Zhang et al., 2023b; Ding et al., 2023; Zhang et al., 2024; Liu et al., 2024c) describe methods to dynamically adapt the rank of different LoRA adapters. Instead of changing the rank, in this work, we propose to dynamically change the scaling of LoRA matrices for different layers, highlighting the need for layer-adaptive methods. PiSSA (Meng et al., 2024) and MiLoRA (Wang et al., 2024) show how improved initialization of LoRA can lead to better performance and faster convergence. Zhu et al. (2024) and Hayou et al. (2024) show that LoRA matrices behave differently in terms of optimal initialization and learning rate. Our work is complementary to these findings, as we also argue for different treatments of LoRAs, but regarding different layers within a model, not within the same adapter. DoRA (Liu et al., 2024a), similarly to our work, targets decoupling of angles and magnitudes, normalizing and scaling the full updated weight matrix $W + \Delta W$ on the column space, controlling each singular column of the finetuned matrices, whereas we propose to normalize the inner $r$-dimensional space of each $\Delta W$ update matrix.

## 5 CONCLUSIONS

In this work, we proposed a novel parameter efficient finetuning method, DeLoRA, which combines the strengths of LoRA –controllable rank– and ETHER –bounded updates– to address their respective limitations. We showed that by normalizing and scaling low-rank updates, DeLoRA is able to effectively decouple the angular learning from the adaptation strength, leading to competitive performance and enhanced robustness. Beyond showing the advantages of DeLoRA, we provided detailed insights into its derivation, from both perspective of LoRA and ETHER, ablating the introduction of each incremental innovation. Finally, we investigated DeLoRA's robustness to learning rate variations and extended training, demonstrating that its decoupled update mechanism is critical for preventing divergence from the pretrained weights. These findings offer valuable perspectives for adapting pretrained models, by addressing key limitations of current PEFT approaches.

## ACKNOWLEDGMENTS

This work was partially funded by the ERC (853489 - DEXIM) and the Alfried Krupp von Bohlen und Halbach Foundation, which we thank for their generous support. The authors gratefully acknowledge the Gauss Centre for Supercomputing e.V. (www.gauss-centre.eu) for funding this project by providing computing time on the GCS Supercomputer JUWELS (Alvarez, 2021) at Jülich Supercomputing Centre (JSC). We would also like to thank Otniel-Bogdan Mercea for the helpful feedback and discussions.

## REPRODUCIBILITY STATEMENT

To facilitate deployment and further research on DeLoRA, we release our implementation code at `https://github.com/ExplainableML/DeLoRA`, as well as ablation studies and hyperparameter choices for all experiments in Appendix B.

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

## A ETHER AND ETHER+ LOW-RANK LIMITATION

In ETHER and ETHER+, even if the applied transformation matrices $I - uu^\intercal$ are full-rank, the resulting weight updates to the pretrained layers are limited to be low-rank. We can show this by rewriting the transformation result in a residual form.

For ETHER the matrix multiplication can be written as:

$$HW = (I - 2uu^\intercal)W$$
$$= W - 2uu^\intercal W$$

where the second term on the right-hand side, by multiplying the pretrained matrix with a rank-1 transformation, restricts the learnable weight updates, which are driven by $u$, to be rank-1.

Similarly, for ETHER+:

$$H^+ W \tilde{H}^+$$
$$= (W - uu^\intercal W + vv^\intercal W)\tilde{H}^+$$
$$= W - uu^\intercal W + vv^\intercal W - (W - uu^\intercal W + vv^\intercal W)\tilde{u}\tilde{u}^\intercal + (W - uu^\intercal W + vv^\intercal W)\tilde{v}\tilde{v}^\intercal$$

where the rank-1 residual matrices on the right-hand side will lead to rank-4 overall weight updates.

This simple mathematical derivation, demonstrates that ETHER and ETHER+ methods are limited to be low-rank, arguably limiting the expressivity and the learning capacity of the two methods.

## B EXPERIMENTAL DETAILS

In this section we report further details about experiments in Section 3, along with hyperparameter choices, and standard deviation results.

**Subject-Driven Generation.** To find the best hyperparameters, we trained and evaluated on the first 3 subjects (10% of the data) for each method among LoRA, DoRA and DeLoRA, all with rank 16. Then, we used best hyperparameters to evaluate each method on all 30 subjects, for 3 different seeds. For LoRA and DoRA we followed best practices and fixed lambda to twice the rank during hyperparemeter search. Optimal learning rate for both methods is 6e-4. For DeLoRA we fixed the $\lambda$ scaling parameter to 1e-3, and found an optimal learning rate of 2e-2 for the $BA$ matrices. Results with standard deviations are reported in Table 6.

| Method | | DINO | CLIP-I |
|---|---|---|---|
| LoRA$_{r=16}$ | (Hu et al., 2022) | $\underline{0.686}_{\pm.0012}$ | $0.818_{\pm.0017}$ |
| DoRA$_{r=16}$ | (Liu et al., 2024a) | $\mathbf{0.687}_{\pm.0015}$ | $\underline{0.819}_{\pm.0015}$ |
| DeLoRA$_{r=16}$ | (ours) | $\underline{0.686}_{\pm.0056}$ | $\mathbf{0.820}_{\pm.0027}$ |

Table 6: Results with standard deviation for subject-driven image generation trained methods. Best scores are highlighted in bold, and second-best scores are underlined.

**GLUE.** Following Wu et al. (2024c), for each benchmark task, we split the publicly available validation set in two subsets as reported in Table 7. When validation sets are larger than 2K, a 1K subset is used as new validation set, and the remaining as test set, otherwise the validation is split in two equally sized subsets. We use the new validation set to tune the hyperparameters on seed 42. Then, best hyperparameters are used to evaluate test performance for seeds 42, 43, 44, 45, 46. For each training run, we use checkpointing to save the best training run, and evaluate with that. For all experiments we use a max sequence length of 512. For larger datasets (MNLI, SST-2, QNLI, QQP) we fix the $\lambda$ scaling learning rate to 3e-3, while for smaller datasets we fix it to 1e-2. For other hyperparameters we run a small grid search. Best values are reported in Table 9. We highlight that with respect to Wu et al. (2024c), we don't discard any underperforming seed. Experiments with standard deviation details are reported in Table 8.

| Splits Sizes | MNLI | SST-2 | MRPC | CoLA | QNLI | QQP | RTE | STS-B |
|---|---|---|---|---|---|---|---|---|
| Training Set | 393K | 67K | 3.7K | 8.5K | 105K | 364K | 2.5K | 5.7K |
| New Validation Set | 1K | 436 | 204 | 522 | 1K | 1K | 139 | 750 |
| New Test Set | 8K | 436 | 204 | 521 | 4.5K | 39K | 138 | 750 |

Table 7: GLUE dataset sizes, with new validation and test splits following Wu et al. (2024c) setup.

| | #param | MNLI | SST-2 | MRPC | CoLA | QNLI | QQP | RTE | STS-B | Avg |
|---|---|---|---|---|---|---|---|---|---|---|
| Full Finet. | 125M | $87.3_{\pm.34}$ | $94.4_{\pm.96}$ | $87.9_{\pm.91}$ | $62.4_{\pm3.29}$ | $92.5_{\pm.22}$ | $91.7_{\pm.19}$ | $78.3_{\pm3.20}$ | $90.6_{\pm.59}$ | 85.6 |
| BitFit | 0.1M | $84.7_{\pm.08}$ | $\mathbf{94.0}_{\pm.87}$ | $88.1_{\pm1.57}$ | $54.0_{\pm3.07}$ | $91.0_{\pm.05}$ | $87.3_{\pm.02}$ | $69.8_{\pm1.51}$ | $89.5_{\pm.35}$ | 82.3 |
| IA3 | 0.06M | $85.4_{\pm-}$ | $93.4_{\pm-}$ | $86.4_{\pm-}$ | $57.8_{\pm-}$ | $91.1_{\pm-}$ | $88.5_{\pm-}$ | $73.5_{\pm-}$ | $88.5_{\pm-}$ | 83.1 |
| LoReFT | 0.02M | $83.1_{\pm.26}$ | $93.4_{\pm.64}$ | $\mathbf{89.2}_{\pm2.62}$ | $60.4_{\pm2.60}$ | $91.2_{\pm.25}$ | $87.4_{\pm.23}$ | $\mathbf{79.0}_{\pm2.76}$ | $90.0_{\pm.29}$ | 84.2 |
| RED | 0.02M | $83.9_{\pm.14}$ | $93.9_{\pm.31}$ | $\mathbf{89.2}_{\pm.98}$ | $61.0_{\pm2.96}$ | $90.7_{\pm.35}$ | $87.2_{\pm.17}$ | $78.0_{\pm2.06}$ | $90.4_{\pm.32}$ | 84.3 |
| LoRA | 0.3M | $86.6_{\pm.23}$ | $93.9_{\pm.49}$ | $88.7_{\pm.76}$ | $59.7_{\pm4.36}$ | $\mathbf{92.6}_{\pm.10}$ | $90.4_{\pm.08}$ | $75.3_{\pm2.79}$ | $90.3_{\pm.54}$ | 84.7 |
| Adapter$^{\text{FFN}}$ | 0.3M | $\mathbf{87.1}_{\pm.10}$ | $93.0_{\pm.05}$ | $88.8_{\pm1.38}$ | $58.5_{\pm1.69}$ | $92.0_{\pm.28}$ | $90.2_{\pm.07}$ | $77.7_{\pm1.93}$ | $90.4_{\pm.31}$ | 84.7 |
| Adapter | 0.4M | $87.0_{\pm.28}$ | $93.3_{\pm.40}$ | $88.4_{\pm1.54}$ | $60.9_{\pm3.09}$ | $92.5_{\pm.02}$ | $\mathbf{90.5}_{\pm.08}$ | $76.5_{\pm2.26}$ | $90.5_{\pm.35}$ | 85.0 |
| DeLoRA(ours) | 0.3M | $86.9_{\pm.21}$ | $93.7_{\pm.79}$ | $88.6_{\pm1.49}$ | $\mathbf{64.7}_{\pm2.33}$ | $\mathbf{92.6}_{\pm.53}$ | $90.2_{\pm.17}$ | $77.3_{\pm1.96}$ | $\mathbf{90.6}_{\pm.38}$ | 85.6 |

Table 8: **GLUE benchmark.** Comparisons of different methods finetuning RoBERTa-base, with standard deviations. Results of all baselines are taken from Wu et al. (2024a) and Wu et al. (2024c).

| Hyperparameters | MNLI | SST-2 | MRPC | CoLA | QNLI | QQP | RTE | STS-B |
|---|---|---|---|---|---|---|---|---|
| $\lambda$ | 12 | 12 | 4 | 4 | 12 | 4 | 12 | 12 |
| Learning Rate | 1e-3 | 1e-3 | 3e-2 | 1e-2 | 3e-3 | 1e-3 | 1e-2 | 1e-2 |
| Batch Size | 32 | 32 | 32 | 8 | 32 | 256 | 8 | 8 |
| Num. Epochs | 30 | 30 | 40 | 80 | 25 | 25 | 80 | 40 |
| Dropout | 0 | 0.1 | 0.2 | 0.2 | 0.25 | 0.25 | 0 | 0.2 |

Table 9: GLUE benchmark hyperparameters.

**Instruction Tuning.** To assess the performance of DeLoRA in finetuning LLMs for Instruction Tuning, we adopted the experimental setup from Bini et al. (2024), finetuning Llama-2-7B (Touvron et al., 2023b) on the Alpaca dataset (Taori et al., 2023) for one epoch, and searching for hyperparameters that deliver the best average performance across MMLU, ARC, and TruthfulQA. For DoRA we used a learning rate of 3e-4, a batch size of 8, and 100 warmup steps. For DeLoRA we used an initial scaling $\lambda$ of 8, learning rates of 1e-2 for $BA$ and 5e-3 for $\lambda$, and other hyperparameters as DoRA. All additional reported results are sourced from Bini et al. (2024).

## C  FIXING THE MAGNITUDE TERM IN DoRA

In the following section we provide preliminary experiments testing if fixing the magnitude in DoRA could lead to similar robustness properties as DeLoRA.

**Performance.** We first evaluate if fixing the magnitude term could be detrimental in terms of performance. Following the setting of our small-scale ablation in Section 3.2, we run a small scale experiment comparing DoRA with its variation.

| Method | DINO | CLIP-I |
|---|---|---|
| DoRA$_{r=16}$(fixed-magnitude) | 0.681 | 0.822 |
| DoRA$_{r=16}$ | 0.683 | 0.820 |

Table 10: Subject-driven Image Generation small-scale ablation

We notice how DoRA results without updating the magnitude term seem to lead to only slightly underperforming results with respect to standard DoRA.

**Robustness.** We then run the same robustness analysis as reported in Fig. 2. We see how fixing the magnitude term does not lead to a behavior similar to DeLoRA, but rather still follows DoRA behavior.

Plots in Fig. 5 show that simply fixing the magnitude term does not alter DoRA robustness properties (Fig. 5, Left), while actually in higher learning rate regimes seems to lead to further divergence (Fig. 5 Right), not allowing the magnitude to counterbalance the divergent trend. This behavior suggests that keeping column norms constant might not be restrictive enough. In this regard, DeLoRA inner normalization in terms of Frobenius distance seems to be a more promising strategy to avoid model divergence.

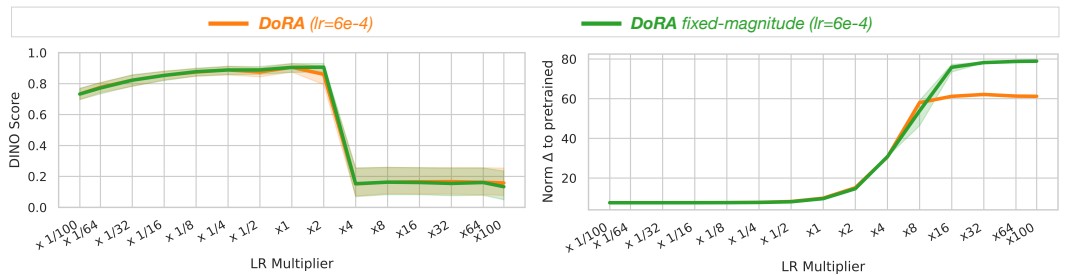

Figure 5: Robustness analysis between DoRA with and without magnitude updates, with respect to learning rate changes from the optimal learning rate.

## D    ROBUSTNESS ABLATION ON DELORA'S BOUNDARY AND ANGLES

We additionally conducted an ablation on DeLoRA's setting, where we run the same robustness analysis of Section 3.4 by varying the learning rate of the scaling term $\lambda$ (affecting the boundary), and the weights $BA$ (angular component). We notice how all methods lead to convergence, additionally demonstrating DeLoRA's robustness properties.

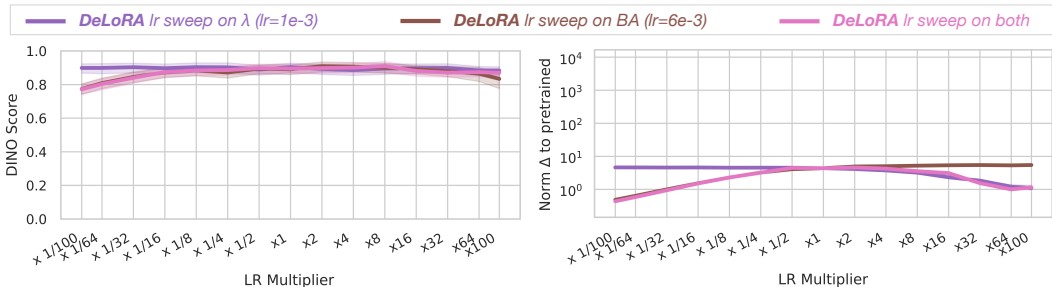

Figure 6: Learning rate robustness plots for DeLoRA in Subject-driven generation task in terms of DINO scores (Left) and Euclidean distance finetuned vs pretrained weights of a projection layer (Right). Ablation testing impact of increasing learning rate for boundary ($\lambda$) or angular weights ($BA$).

# E QUALITATIVE EXAMPLES

We report in Fig. 7 qualitative examples generated by our propoosed DeLORA finetuning Stable Diffusion for the tasks of Subject-driven Generation and Semantic Map to Image. While in Figure 8 we report qualitative examples of prolonged genearation with DeLoRA, LoRA and DoRA methods.

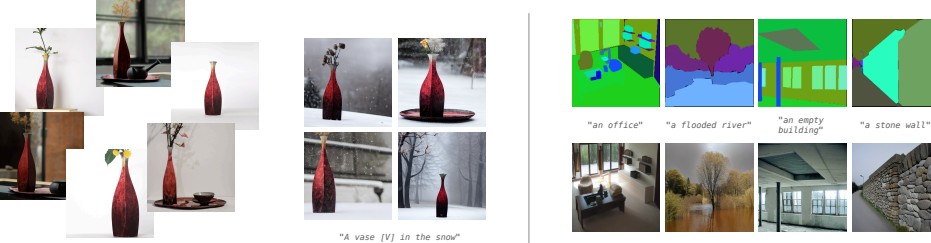

Figure 7: Examples generated by DeLoRA-finetuned Stable Diffusion for personalized generation on a small set of subject-specific images (left), and for semantic map to image on ADE20K (right).

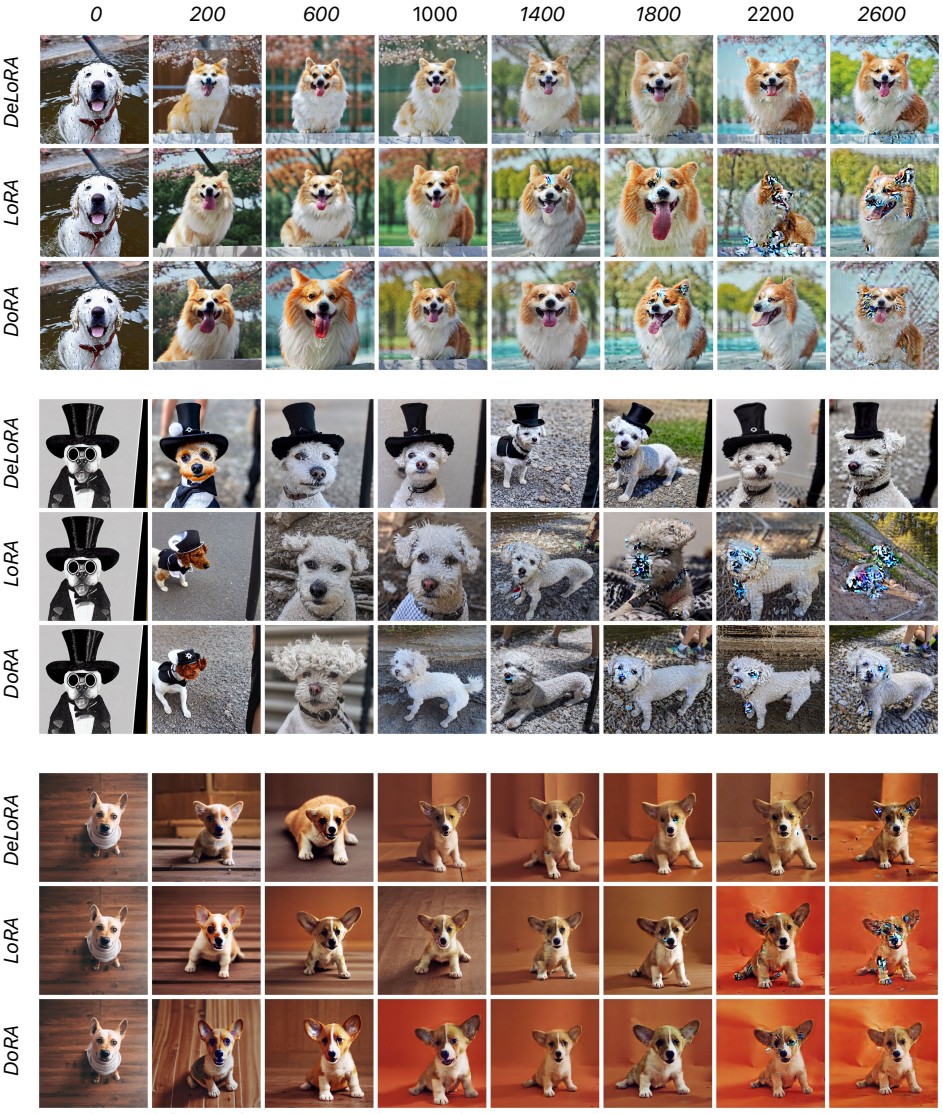

Figure 8: Prolonged finetuning generated examples generated by DeLoRA, LoRA, and DoRA methods, up to time step 2600.

