# OpenReview forum: "Decoupling Angles and Strength in Low-rank Adaptation"
_ICLR.cc/2025/Conference — ICLR 2025 Poster_

### Official Review · Reviewer_EKmU · 2024-10-27

**Soundness:** 2
**Presentation:** 3
**Contribution:** 2
**Rating:** 6
**Confidence:** 3

**Summary:**

Motivated by the lack of robustness of LoRA fine-tuning with respect to hyperparameter choice and catastrophic forgetting phenomena, the authors propose a novel PEFT method trying to tackle these problems. Their proposal merges the advantages of LoRA and ETHER, being able thus to control rank and maximal distance between fine-tuned and frozen weights.

**Strengths:**

The paper is well presented and the proposed methods is both efficient and easy to implement.

**Weaknesses:**

While I find no evident weakness in the manuscript and its content, I believe a broader experimental setting should be proposed to test the method (see questions for more details). Moreover, given the motivation of the work, I encourage the authors to include additional evidence concerning the robustness of their proposed method with respect to hyperparameters (and maybe something more about the avoidance of forgetting phenomena).

**Questions:**

I leave here some questions/comments concerning the work:
1) Would it be possible to test the method on other common benchmarks (e.g. GLUE Deberta V3)?
2) The only concern I have about the new parametrization is that if a column of $A$ and $B$ is driven towards zero during training, normalization may cause instabilities. Did the authors ever observe this in practice? I would be worried of this phenomenon happening especially when choosing a high rank for the correction.

---

> ### Author Response · Authors · 2024-11-25
> **Authors Response to Reviewer EKmU**
>
> We thank the reviewer for their feedback, and provide our response to each raised issue separately below.
>
> __[Wa - Q1] Lack of broader experimental setting (e.g. GLUE on DeBERTa-v3)__
>
> We thank the reviewer for the benchmark suggestion. In the current manuscript, we tested how DeLoRA performs on the GLUE benchmark. Rather than DeBERTa-v3, we decided to finetune RoBERTa-base for time/resources reasonings, and in order to compare in the same setting of [Wu Z et al., Wu M et al.], which we find to be fairer than usually reported benchmarks on the validation sets.
>
> For details about the GLUE benchmark, we refer to the Global Authors Response above, and to the updated manuscript.
>
> __[Wb] Encourage the authors to include additional robustness evidence (and other analysis)__
>
> We thank the reviewer for the suggestion. As highlighted by reviewer pmk9, we notice that we omitted to describe Fig.2 (previously Fig.4) in the manuscript.
>
> We updated the manuscript with proper description, and further updated Fig.2 (previously Fig.4) introducing analysis on the weights updates (Fig.2, Right).
>
> In Appendix D, we also provide a further ablation demonstrating DeLoRA robustness to learning rate changes on angular and scale components (Fig.6). We agree that further analysis, for example in continual learning settings, could be an interesting future research direction
>
> __[Q2] Could it happen that a column of A and B is driven towards zero during training, causing instabilities?__
>
> When testing different initialization schemes, we indeed found that initializing B with zeros (with tolerance to avoid explosion) would sometimes lead to instabilities. In our final scheme where we initially subtract a copy of BΞA from the original weights instead, we didn’t see this happening in practice. We notice that to potentially avoid instabilities issues, we did not use weight decay.
>
> References:
>
> [Wu Z et al.] Zhengxuan Wu, Aryaman Arora, Zheng Wang, Atticus Geiger, Dan Jurafsky, Christopher D. Manning, and Christopher Potts. Reft: Representation finetuning for language models, 2024b
>
> [Wu M et al.] Muling Wu, Wenhao Liu, Xiaohua Wang, Tianlong Li, Changze Lv, Zixuan Ling, Jianhao Zhu, Cenyuan Zhang, Xiaoqing Zheng, and Xuanjing Huang. Advancing parameter efficiency in finetuning via representation editing, 2024a.

---

> > ### Comment · Reviewer_EKmU · 2024-11-25
> >
> > I wish to thank the reviewers for carefully addressing my concerns.
> > I also read the other reviews and I am satisfied with the new results and the answers, therefore I am gonna raise my score.
> > As a final comment, I suggest the authors to add a small discussion about the initialization in the main manuscript.

---

> > > ### Author Response · Authors · 2024-11-26
> > > **Authors Response to "Official Comment by Reviewer EKmU"**
> > >
> > > We thank the reviewer for the feedback and for the score increase.
> > >
> > > Following the reviewer’s suggestion we updated the manuscript to extend the initialization description (lines 166-176). We report here the updated section:
> > >
> > > “To initialize the finetuning process from the pretrained model, DeLoRA's normalization operation does not allow to simply follow LoRA's zero initialization of the B matrix. From preliminary experiments, we find that introducing a small epsilon to avoid division by 0, would sometimes lead to unstable results. Therefore, we instead take inspiration from (Bini et al.) and subtract a copy of the kaiming-randomly initialized B and A matrices. With respect to(Bini et al.), we simply freeze these additional parameters and merge them to the pretrained weights, as in
> > >
> > > $W=\bar{W} - \frac{λ||W||}{r}(BΞA)_0$
> > >
> > > where $\bar{W}$ is the original pretrained matrix, and $\frac{λ||W||}{r}(BΞA)_0$ is the update matrix at time 0. With this modification, no significant computational overhead is introduced.”

---

> > > > ### Comment · Reviewer_EKmU · 2024-11-30
> > > >
> > > > I thank the authors again for the response. I am satisfied with the rebuttal and I am gonna keep my score.

---

### Official Review · Reviewer_vsop · 2024-11-01

**Soundness:** 3
**Presentation:** 3
**Contribution:** 2
**Rating:** 3
**Confidence:** 5

**Summary:**

This paper addresses some limitations in current parameter-efficient fine-tuning (PEFT) methods, particularly the robustness and flexibility challenges faced by additive methods like LoRA and multiplicative approaches like ETHER. The authors introduce DeLoRA, a fine-tuning method that normalizes and scales learnable low-rank matrices to improve hyperparameter robustness without sacrificing performance. Through evaluations on subject-driven image generation and large language model (LLM) instruction tuning tasks, DeLoRA demonstrates consistent improvements over existing PEFT methods, offering a balanced solution that enhances adaptability and performance in fine-tuning applications.

**Strengths:**

1.	Overall, this paper is well-written and presents its ideas in a clear, accessible manner, making it easy to follow.
2.	The authors provide insightful reviews of two primary categories of fine-tuning methods: additive techniques such as LoRA and multiplicative approaches like ETHER, effectively highlighting the limitations inherent to each approach.
3.	The bounded approach constrains the updating scheme within a Frobenius norm ball, promoting a more robust selection of learning rates and potentially enhancing stability across different training settings.

**Weaknesses:**

Overall, the contribution of this paper appears somewhat incremental, as it focuses on constraining weight updates within a restricted range—a concept that has also been examined in the DoRA paper. Additionally, the performance comparison with LoRA and DoRA is challenging to assess directly, as the experiments were conducted on different datasets.

1.	[Key Issue] The concept of decoupling the angles and scales in LoRA has been explored extensively in the DoRA method. Consequently, the contribution of this paper feels somewhat incremental in comparison to DoRA. Similar to DeLoRA, DoRA can limit its scale term within a fixed range, thereby preventing over-drifting in downstream adaptations. Would this lead to the same robustness effect?
2.	[key issue] LoRA and DoRA algorithms are typically evaluated on well-established NLP and NLU tasks, such as the GLUE benchmarks and Commonsense Reasoning, where their performance characteristics are widely understood. Thus, a fairer performance comparison would be achievable if DeLoRA were also tested on these datasets.
3.	In Eq (13), the updating is limited to |H-I| <= r, effectively limiting weight changes to within a Frobenius norm ball. Then it is not surprising that the algorithm is not sensitive to the change of learning rates. But a fundamental question is: is this restriction always necessary? For instance, in cases where the pretrained model diverges significantly from the target task, imposing a strict parameter update range might inadvertently hamper overall performance.
4.	In line 172, the authors state that 'multiplicative fine-tuning methods show stronger performance in generative models.' However, the results in Table 2 suggest that additive methods like LoRA and DoRA outperform OFT approaches, which appears contradictory and requires clarification.
5.	In Table 2, the performance improvement over DoRA appears marginal. Could the authors clarify if DoRA was evaluated with r=16? To ensure statistical reliability, it would be beneficial to repeat this experiment multiple times and report the standard deviation, as the observed improvement may fall within the range of experimental error.

**Questions:**

See the above "weakness".

---

> ### Author Response · Authors · 2024-11-25
> **Authors Response to Reviewer vsop [1/2]**
>
> We thank the reviewer for their feedback, and provide our response to each raised issue separately below.
>
> __[W1a] Contribution feels somewhat incremental in comparison to DoRA.__
>
> We thank the reviewer for the raised concern, we provide a more detailed analysis on the differences with DoRA in our Global Authors Response above, and we added a paragraph in Section 2.2 of the manuscript to better clarify the differences.
>
> __[W1b] Similar to DeLoRA, DoRA can limit its scale term within a fixed range, thereby preventing over-drifting in downstream adaptations. Would this lead to the same robustness effect?__
>
> With DeLoRA, in order to let the boundary to optimally adapt to different layers, the λ parameter is learnable, making its boundaries equivalent to the current DoRA implementation (being unbounded, in principle).
>
> At the same time, as described in our comparison with DoRA in our Global Authors Response above, divergence does not happen. This suggests that our proposed method is able to decouple the transformation strength from the angular learning more effectively than DoRA, leading to the robustness properties as shown in Fig.2 (previously Fig.4).
>
> To provide a more concrete test, we decide to test how DoRA would perform while fixing its magnitude term: if this would lead to robustness properties similar to DeLoRA. Experiments and preliminary results are reported in Appendix C in our updated manuscript.
>
> Summarizing our findings, we test DoRA with fixed magnitude (i)  on performance, where it shows to be only slightly underperforming w.r.t. the original DoRA, and (ii) on robustness, where plots between the two DoRA variations closely overlap.
>
> This behavior suggests that keeping column norms constant might not be restrictive enough, whereas DeLoRA inner normalization seems to be more effecting to avoid model divergence.
>
> __[W2a] LoRA and DoRA algorithms are typically evaluated on well-established NLP and NLU tasks, such as the GLUE benchmarks and Commonsense Reasoning__
>
> We thank the reviewer for the suggestion. We now tested our proposed DeLoRA method on the GLUE benchmark. We refer to the Shared Response for the details.  For time/resources reasons we are not able to evaluate on the Commonsense Reasoning benchmark.
>
> Given the same pretrained network (Llama2-7B), we argue our Instruction Tuning benchmark could be sufficient to show DeLoRA abilities in finetuning large-scale Language Models
>
> __[W2b] Is the Frobenius ball restriction always necessary? There might be cases where the pretrained model diverges significantly from the target task__
>
> We agree that there might be tasks that might require stronger divergence from the pretrained network.
> With our DeLoRA proposal, based on the domain of the target task, one can arbitrarily control this ball restriction by initially tuning λ.
>
> In Fig. 2 (and Fig.6 in Appendix D) of our updated manuscript, we demonstrate that using high learning rates for the boundary does not lead to divergence. This suggests that our proposed method can effectively adapt to domains where the optimal ball variance differs significantly across layers.
>
> We believe exploring task-specific lambda values is a promising future direction, particularly in determining how these values should be adjusted based on the similarity between the current task and the original pretraining domain.

---

> > ### Author Response · Authors · 2024-11-25
> > **Authors Response to Reviewer vsop [2/2]**
> >
> > __[W3] authors state that multiplicative fine-tuning methods show stronger performance in image generative models, while results in Table 2 suggest that additive methods like LoRA and DoRA outperform OFT approaches, which appears contradictory and requires clarification__
> >
> > [Qiu et al.] and [Liu et al.]’s findings suggest that multiplicative approaches tend to outperform additive approaches (specifically in Subject-driven Generation and Semantic Map to Image benchmarks). However indeed the reported OFT result from [Bini et al.] does not show stronger results.
> >
> > On the other hand, we reported a more in depth analysis in our ablation experiments (Tab.1,2), where we find that the multiplicative derivation “ETHER+ + controllable scale + high rank + relaxed” outperforms the “LoRA + normalize w/ controllable boundary” method in both DINO (0.696 vs 0.682) and CLIP-I (0.833 vs 0.809).
> >
> > Therefore we assume that a reason for this might be the layer-aware finetuning strength, which we introduce with our “weights-norm scaling” proposal
> >
> > __[W4] Is DoRA in Table 2 reported for rank r=16?  In addition, it would be beneficial to repeat the experiments multiple times and report the standard deviation.__
> >
> > We thank the reviewer for noting our oversight regarding the rank specification in the DoRA experiments. We used rank=16 to ensure a fair comparison between methods with comparable parameter counts.
> >
> > We have expanded our analysis by running the Dreambooth benchmark across three different seeds. In Appendix B, we provide detailed documentation of our experimental setup to ensure reproducibility, along with the corresponding standard deviations of our results.
> >
> > Results show that LoRA, DoRA and DeLoRA are indeed comparable in performance (DINO scores of 0.686, 0.688, 0.687, and CLIP-I scores of 0.818, 0.819, 0.820, respectively). We highlight anyhow DeLoRA’s robustness advantage over the alternatives, which would make DeLoRA the favorite alternative.
> >
> > References:
> >
> > [Qiu et al.] Zeju Qiu, Weiyang Liu, Haiwen Feng, Yuxuan Xue, Yao Feng, Zhen Liu, Dan Zhang, Adrian Weller, and Bernhard Sch¨olkopf. Controlling text-to-image diffusion by orthogonal finetuning. In NeurIPS, 2023
> >
> > [Liu et al.] Weiyang Liu, Zeju Qiu, Yao Feng, Yuliang Xiu, Yuxuan Xue, Longhui Yu, Haiwen Feng, Zhen Liu, Juyeon Heo, Songyou Peng, Yandong Wen, Michael J. Black, Adrian Weller, and Bernhard Sch¨olkopf. Parameter-efficient orthogonal finetuning via butterfly factorization. In ICLR, 2024b.
> >
> > [Bini et al.] Massimo Bini, Karsten Roth, Zeynep Akata, and Anna Khoreva. ETHER: Efficient finetuning of large-scale models with hyperplane reflections. In ICML, 2024.

---

> > > ### Comment · Reviewer_vsop · 2024-11-26
> > > **Degraded Performance for your Table**
> > >
> > > Thanks for your detailed response.
> > > 1) I would agree that a trainable lambda would be necessary.
> > > 2) The reviewer observes that the GLUE results presented in your paper are significantly lower than those reported in the original work. The reviewer has some hands-on experience that LoRA performance should be higher than what you report. Can you clarify?

---

> > > > ### Author Response · Authors · 2024-11-26
> > > > **Authors Reply to "Degraded Performance for your Table"**
> > > >
> > > > The reason for the LoRA reported results being lower than in the original work come for the evaluation procedure being used.
> > > >
> > > > [Wu M. et al.] in their ACL24 work, recently pointed out two issues in previous evaluation settings of PEFT methods for the GLUE benchmark: (1) MRPC, RTE, and STS-B tasks are evaluated after pre-finetuning on MNLI dataset, which introduces complexity to the pipeline, and more importantly (2), since there is no publicly available test set, evaluations are performed on the validation set, which is also used as test set (as also confirmed by LoRA’s main author in his reply to a github issue here https://github.com/microsoft/LoRA/issues/31), violating common best practice that the test set should not influence model selection.
> > > >
> > > > For this reason, we follow [Wu M. et al.] procedure instead, as also adopted by [Wu Z. et al.]. For each benchmark task, we split the publicly available validation set into two subsets. We then use one subset to tune the hyperparameters on seed 42, and use best hyperparameters to evaluate test performance for seeds 42, 43, 44, 45, 46 (further details in our revised manuscript, lines 358-406 and 738-742). LoRA results are reported by [Wu M. et al.], and are expected to be lower to the original paper’s ones.
> > > >
> > > > References:
> > > >
> > > > [Wu M et al.] Muling Wu, Wenhao Liu, Xiaohua Wang, Tianlong Li, Changze Lv, Zixuan Ling, Jianhao Zhu, Cenyuan Zhang, Xiaoqing Zheng, and Xuanjing Huang. Advancing parameter efficiency in finetuning via representation editing. In ACL 2024.
> > > >
> > > > [Wu Z et al.] Zhengxuan Wu, Aryaman Arora, Zheng Wang, Atticus Geiger, Dan Jurafsky, Christopher D. Manning, and Christopher Potts. Reft: Representation finetuning for language models. In NeurIPS 2024.

---

> > > > > ### Comment · Reviewer_vsop · 2024-11-27
> > > > >
> > > > > Thanks for your update, and I have raised my score.

---

### Official Review · Reviewer_pmk9 · 2024-11-03

**Soundness:** 3
**Presentation:** 3
**Contribution:** 2
**Rating:** 6
**Confidence:** 3

**Summary:**

This paper proposes a new PEFT method based on LoRA called DeLoRA.
In specific, the authors propose to decompose the learned low-rank update in LoRA into a learnable low-rank matrix with a unit norm, and a learnable magnitude.
The authors also draw a connection between the proposed DeLoRA and multiplicative ETHER which learns low-rank matrices to be *multiplied* instead of *added to* the original frozen parameters.
The authors evaluated the proposed method on a variety of language and vision tasks and demonstrated fine-tuning performance improvement from DeLoRA.

**Strengths:**

1. The writing of this paper is good and it is pleasant to read.

2. The proposed method is well-presented. The connection to existing methods ETHER and DoRA is also amenable.

3. The experiments look sufficient to me.

**Weaknesses:**

After reading through the paper, my main concerns lie in the concept of DeLoRA. Namely,

1. While an original main motivation of DeLoRA, as presented in Introduction, is to "introduce a boundary on the weight updates" to LoRA, the addition of *learnable* parameter $\lambda$ (without any regularization/constraint, if I read the paper correctly) removes this guarantee. As a result, learnable DeLoRA solution is just a strict subset of LoRA's --- the proposed parameterization introduces some inductive bias. Its underlying rationale, which cannot be fully explained by the presented motivation, is unclear.

2. On the other hand, as the idea of decomposing the learning into magnitude and direction has already been proposed in DoRA, I feel that how different DeLoRA really is from DoRA, e.g., if they are in fact equivalent, needs more elaboration and careful discussion. Currently the authors only provide a equation-by-equation comparison, but no insights is provided in this form.
In addition, I think this difference should be highlighted in Introduction and Method, instead of just in Related Work.

**Questions:**

Besides above questions in Weakness, I also wonder:

1. In Table 1,2 how was controllable boundary combined with LoRA?

2. In Fig 3, DeLoRA's distance to initialization was in fact decreased, how was this happened?

3. Can you elaborate more on Fig 4? I didn't find any discussion in the main body.

4. Can you test the proposed method on more language tasks, such as math reasoning?

I am willing to adjust my score if these questions can be addressed.

---

> ### Author Response · Authors · 2024-11-25
> **Authors Response to Reviewer pmk9**
>
> We thank the reviewer for their feedback, and provide our response to each raised issue separately below.
>
> __[W1] In DeLoRA, treating the boundary λ as a learnable parameter will remove the guarantee of non-divergence etc, making DeLoRA a special case of LoRA__
>
> Since DeLoRA's weight updates are rank-r and impose stricter Euclidean distance constraints, its potential update space necessarily forms a subset of LoRA's update space.
>
> However, in DeLoRA, by decoupling the update strength (with learnable parameter λ) from the angular learning (BA matrices), the resulting training dynamics lead to extremely different behaviors, as shown when evaluating Frobenius distances in Fig.3 (left).
> Plot in Fig.3 shows that LoRA and DoRA updates exhibit near-linear growth with each iteration, whereas DeLoRA updates rapidly approach the boundary value before gradually decreasing.
>
> At the same time, while having a learnable boundary λ might seem to potentially lead to the same divergence issues of LoRA, thanks to DeLoRA’s decoupling this does not happen. This can be seen in the robustness plots in Fig.2 (previously Fig.4), where DeLoRA maintains optimal performance at learning rates up to 6e-1, while LoRA's performance degrades at just 2.4e-3.
>
> In our updated manuscript we include an additional analysis on the Frobenius distance of a projection layer in an attention module at convergence, with respect to its pretrained counterpart (Fig.2, Right). These measurements reveal that while LoRA and DoRA diverge from the initial weights, DeLoRA remains stable despite having the capacity for larger deviations.
>
> We further analyze DeLoRA’s robustness separating variations on boundary (λ0 and anglular components (BA), further demonstrating its robustness (Fig.6, Appendix D).
>
> __[W2] Given the similarities with DoRA, the differences should be stressed more. Plus, comparison with DoRA should be moved to Method section.__
>
> We thank the reviewer for the suggestions. We addressed this issue in the Global Authors Response above.
>
> __[Q1] In Table 1,2 how was the controllable boundary combined with LoRA?__
>
> The “normalization + controllable boundary” refers to the addition of the normalization matrix Ξ and of the scale λ to LoRA, which will determine the upper bound on the weight updates in terms of Frobenius distance. We refer to the boundary as controllable because its initialization can be arbitrarily tuned, to better fit the task at hand.
>
> To improve clarity, in our ablations tables (Tab.1,2)  we added a column describing all different formulations.
>
> __[Q2] In Fig 3, DeLoRA's distance to initialization was in fact decreased, how was this happened?__
>
> As derived from Section 2.2, Eq. 7, the λ parameter determines an upper bound, allowing the distance of the weight updates to be lower than the maximum value. Interestingly, DeLoRA finetuning dynamics show that the updates’ distance initially gets close the boundary and then, as training continues, it tends to decreases, significantly different behavior w.r.t. LoRA and DoRA dynamics.
>
> __[Q3] Can you elaborate more on Fig 4? I didn't find any discussion in the main body.__
>
> We thank the reviewer's observation regarding Fig.4 (now Fig.2), which we had inadvertently omitted from the manuscript. We have added a detailed description of the updated figure in Section 3.4 of the updated manuscript.
>
> We report below here its description:
>
> “We conducted a comprehensive learning rate robustness analysis in the setting of the Subject-driven Generation task of Section 3. Evaluation is done reporting DINO scores (Fig.2, Top) and Euclidean distance between finetuned and pretrained weights of an inner layer (Fig.2, Bottom) across multiple methods, using a range of learning rates derived from each method’s base learning rate (Fig.2, Left).
> Our analysis shows that DeLoRA is able to achieve the same robustness of ETHER+, while improving performance, while both LoRA and DoRA performance degrade at 4× the base learning rate. We also notice how LoRA updates’ distance grows at higher learning rates, while interestingly DoRA, after 8×, does not diverge further, likely due to its magnitude control. However this does not lead to better performance in these regimes.”
>
> __[Q4] Can you test the proposed method on more language tasks, such as math reasoning?__
>
> As suggested by multiple reviewers, we have now evaluated our DeLoRA method in finetuning for Natural Language Understanding on on the GLUE benchmark. We discussed this additional benchmark in the Global Authors Response above.

---

> ### Author Response · Authors · 2024-11-30
>
> As the end of the discussion period approaches, we again thank the reviewer for their valuable feedback, which helped us improve our manuscript, and hope that our answers positively addressed the reviewer’s concerns: (__W1__) by clarifying how empirically DeLoRA does not lead to divergence, supported by new experiments and analysis, (__W2__) by extensively discussing differences with DoRA, also testing how DeLoRA achieves better decoupling with new experiment fixing DoRA’s magnitude, in Appendix C, and (__Q4__) by providing new results on the GLUE natural language understanding benchmark..
> Plus, we hope to have fully answered reviewer’s questions (__Q1__, __Q2__, __Q3__), which helped us improve our manuscript by clarifying our ablation studies with the addition of mathematical formulations in Tab.1,2, and by creating a separate section for DeLoRA’s robustness analysis.
>
> We take the chance to ask the reviewer if they have any further concerns, giving us the chance to address these too, and hope that our answers and the overall manuscript improvements were satisfactory. If so, we hope that the reviewer would consider raising their score.

---

> > ### Comment · Reviewer_pmk9 · 2024-11-30
> >
> > Dear authors,  Thanks for your detailed rebuttal. In response, I raised my score accordingly.

---

### Official Review · Reviewer_nYwf · 2024-11-03

**Soundness:** 3
**Presentation:** 2
**Contribution:** 2
**Rating:** 6
**Confidence:** 4

**Summary:**

Two existing fine-tuning methods, LoRA and ETHER, have shown widespread success across the field, especially the former. However, some issues with LoRA from hyperparameter tuning and length of fine-tuning arise, leading to degradation in performance. The proposed method, DeLoRA, separates the magnitude and direction of the weight updates from LoRA. This is reminiscent of multiplicative weight updates such as ETHER and OFT with constrained magnitudes. DeLoRA is shown empirically to frequently achieve better accuracy than either LoRA or ETHER.

**Strengths:**

- The ablation studies provide a convincing argument that somewhere in the continuum between LoRA and ETHER+ is a more powerful method. The experiments showing that different settings have different optimal locations in this continuum show that there are some directions forward with regard to which settings require different method choices, as well as defend that this continuum often contains a better method than the extremes.

- DeLoRA shows promise in having strong robustness similar to ETHER with explicit weight constraints, even when learning rates are high.

**Weaknesses:**

- It took some time to figure out what exactly the DeLoRA method is. For clarity, what DeLoRA is could be written in its own dedicated section, not just embedded in Figure 1 without explaining the parameters or at the end of Section 2.2.1, since the two equations describing the method seem like all the others in the derivation at first glance.

- It's mentioned in the introduction that one downside to LoRA is performance degradation during extended fine-tuning. This reads as though DeLoRA will overcome this issue, however, the top right of Figure 3 seems to show DeLoRA having the same issue. Are there other instances that can be added to show that DeLoRA does robustly train over many iterations?

- Figure 3 could use some repetitions. The single training run makes some of the training dynamics, especially DoRA's, hard to trust that they weren't due to randomness.

**Questions:**

- In the derivations section, the LoRA and ETHER derivations seem to create different methods. This may come from the step of changing the ETHER derivation from a multiplicative adaptation to an additive one, but how this is achieved is not clear. Adding at least a few more steps to bridge between the two would be appreciated.

Small things:
- "Figure 5: Average..." instead of "Figure 5: Avergae..."

---

> ### Author Response · Authors · 2024-11-25
> **Authors Response to Reviewer nYwf**
>
> We thank the reviewer for their feedback, and provide our response to each raised issue separately below.
>
> __[W1] For clarity the DeLoRA method could have its own section__
>
> We thank the reviewer for the suggestion. In our revision, we added a paragraph to the updated manuscript in section 2.2 named “DeLoRA formulation” summarizing our DeLoRA setup and key elements
>
> __[W2-3] From Fig.3, DINO scores for DeLORA seem to degrade over time, plus current plot would gain if averaged over multiple runs__
>
> Indeed, DeLoRA performance degrade over time, however qualitative examples in Fig.3 (bottom) and in Appendix show that this degradation happens slower than for LoRA (artifacts for DeLoRA slightly appear at iteration 1200, and strongly appear at iteration 2400, while for LoRA artifacts are already strong from iteration 900).
>
> In this regard, we agree with the reviewer that DINO scores for prolonged finetuning as reported in Fig.3 are confusing. In this regard, we highlight that results with high distortion/artifacts can still show good DINO score (e.g. LoRA finetuned model at iteration 1500, which reaches a DINO score above 0.7, while showing a strong presence of artifacts).
>
> For this reason, we decide to remove the DINO score plot in the updated version of the manuscript, and decide to only compare different methods on qualitative examples. For a fair evaluation, we include additional randomly sampled qualitative examples of prolonged generation in Appendix E.
>
> __[Q1] Derivation of DeLoRA from ETHER, especially the step from multiplicative to additive FT, is not clear__
>
> The transition from multiplicative to additive finetuning transforms the multiplicative update W' = (I - λ/r(BΞΑ - DΦC))W with weight change ΔW = λ/r(BΞΑ - DΦC)W into DeLoRA's additive form W' = W + λ||W||/r(BΞΑ - DΦC), where DΦC remains frozen at BΞΑ's initial value. To facilitate comparison, we added a column in Tables 1 and 2 that details these different formulations.

---

> > ### Author Response · Authors · 2024-11-30
> >
> > As the end of the discussion period approaches, we again thank the reviewer for their valuable feedback, and hope their concerns have been positively addressed: (__W1__) by reorganizing our method into a dedicated section to improve readability and clarity, (__W2__, __W3__) by re-discussing the prolonged finetuning results and presentation, adding new qualitative examples in Appendix E, and (__Q1__) by clarifying our ablation studies, also adding mathematical formulations to Tab.1,2.
> >
> > We also want to point out that thanks to all reviewers’ feedbacks and suggestions, our final manuscript has significantly improved in _experimental evidence_ (new benchmark results), _analysis_ (comparisons with DoRA, robustness analysis, and further evaluations), and _clarity_ (separate sections for: DeLoRA summary, DoRA comparison, robustness analysis), recommending an overview of the overall manuscript improvements.
> >
> > We take the chance to ask the reviewer if they have any further concerns, giving us the chance to address these too, and hope that our answers were satisfactory. If so, we hope that the reviewer would consider raising their score.

---

### Author Response · Authors · 2024-11-25
**Global Authors Response [1/3]**

We would like to thank all the reviewers for the detailed and useful feedback.

We appreciate the acknowledgement for our proposed robust parameter-efficient finetuning (PEFT) method (nYwf, pmk9, vsop, EKmU), alongside its derivation from LoRA and ETHER methods (pmk9, nYwf, EKmU) and our proposal for solving their limitations (nYwf, vsop), while highlighting that the paper is well presented and easy to read (pmk9, vsop, EKmU).

For common questions and concerns we reply below here, while for each individual discussion point we refer to their respective response below. We also modify the main manuscript, highlighting in blue the relevant changes.

---

> ### Author Response · Authors · 2024-11-25
> **Global Authors Response [2/3]**
>
> __[W- pmk9,vsop,EKmU] Paper would benefit from evaluating on GLUE benchmark (vsop, EKmU, pmk9).__
>
> We thank the reviewers for the benchmark recommendation. We now evaluated our proposed method by finetuning RoBERTa-base on the GLUE benchmark following [Wu M. et al.] setting, which was proposed to fairly evaluate GLUE on the publicly available validation set following best practices. Results are reported below (Table 4 in the updated version of the manuscript), while corresponding standard deviations and best hyperparameters are reported in Appendix B). Results for all baselines come from [Wu M. et al., Wu Z. et al.]. Our method achieves superior performance on CoLA (64.7%), QNLI, and STS-B, and attains a significantly higher average score that matches full finetuning results.
>
>
> |	        |#params|MNLI|	SST-2|	MRPC|	CoLA|	QNLI|	QQP|	RTE|	STS-B|	Avg|
> |----------|-----------|------|-------------|------------|--------|----------|----------|----------|----------|--------|
> Full Finet.|	125 M|	87.3|	94.4	|87.9	|62.4	|92.5	|91.7	|78.3|	90.6|	85.6|
> BitFit	| 0.1 M	|84.7	|__94.0__	|88.1	|54.0	|91.0	|87.3	|69.8	|89.5	|82.3|
> IA3	         |  0.06 M|	85.4	|93.4	|86.4	|57.8	|91.1	|88.5	|73.5	|88.5|	83.1|
> LoReFT	|0.02 M|	83.1	|93.4	|__89.2__	|60.4	|91.2	|87.4	|__79.0__	|90.0	|84.2|
> RED	|0.02 M	|83.9	|93.9	|__89.2__	|61.0	|90.7	|87.2	|78.0	|90.4	|84.3|
> LoRA	|0.3 M|	86.6	|93.9	|88.7	|59.7	|__92.6__	|90.4	|75.3	|90.3	|84.7|
> Adapter(FFN)|0.3M |__87.1__	|93.0	|88.8	|58.5	|92.0	|90.2	|77.7	|90.4	|84.7|
> Adapter	|0.4 M|	87.0	|93.3	|88.4	|60.9	|92.5	|__90.5__	|76.5	|90.5	|85.0|
> DeLoRA(ours)| 0.3 M |86.9	|93.7	|88.6	|__64.7__	|__92.6__	|90.2	|77.3	|__90.6__	| __85.6__ |
>
>
> Experimental details: by following [Wu Z. et al., Wu M. et al.], we split the validation data in two subsets, and tune the hyperparameters on the smaller subset with seed 42. Then, we use the best validation hyperparameters, re-train on the training data for seeds {42,43,44,45, 46}, and get test results on the larger validation split. In addition, to be consistent with LoRA reported results, we only apply DeLoRA (rank 8) to Q and V matrices, which is likely underperforming with respect to applying lower-rank DeLoRA modules to more linear layer types (so with comparable number of parameters) [Hu et al.2022]
>
> References:
>
> [Wu M. et al.] Muling Wu, Wenhao Liu, Xiaohua Wang, Tianlong Li, Changze Lv, Zixuan Ling, Jianhao Zhu, Cenyuan Zhang, Xiaoqing Zheng, and Xuanjing Huang. Advancing parameter efficiency in finetuning via representation editing, 2024a
>
> [Wu Z. et al.] Zhengxuan Wu, Aryaman Arora, Zheng Wang, Atticus Geiger, Dan Jurafsky, Christopher D. Manning, and Christopher Potts. Reft: Representation finetuning for language models, 2024b
>
> [Hu et al.2022] Edward J Hu, Phillip Wallis, Zeyuan Allen-Zhu, Yuanzhi Li, Shean Wang, Lu Wang, Weizhu Chen, et al. Lora: Low-rank adaptation of large language models. In ICLR, 2022.

---

> ### Author Response · Authors · 2024-11-25
> **Global Authors Response [3/3]**
>
> __[W- vsop, pmk9] Given the similarities with DoRA, the differences should be stressed more (vsop, pmk9)__
>
> We thank the reviewers for the suggestion. We agree that a more detailed comparison with DoRA could be beneficial to better highlight the substantial differences, given that both methods address the finetuning process with similar premises, i.e. by trying to decouple the learning of magnitudes and angles.
>
> Following reviewer pmk9's suggestion, we added a new paragraph to the Methods section that clarifies the fundamental differences between the approaches, particularly in their implementation of normalization and scaling, which results in substantially distinct properties for each method.
>
> We report the paragraph below, and refer for Figures to the updated manuscript:
>
> “DoRA (Liu et al., 2024a), similarly to our work, addresses the finetuning process by trying to decouple the learning of magnitudes and angles, by using a formulation that leads to weight updates W ′ = m (W +∆W)/||W +∆W||.
>
> We can summarize the key differences of DoRA with respect to our proposal into two: (i) the normalization and scaling operations happen on the fully finetuned weights, and (ii) these operations happen on the column space of the weight matrices, which draw a significant difference to our proposal.
>
> We argue that DeLoRA finetuning (i) by introducing the normalization and scaling operations directly on the weight updates ∆W , it more directly tackles the goal of not diverging from the pretrained model, and (ii) by normalizing the inner low-dimensional space (rather than the column space), it actually results in an implicit Frobenius-distance boundary, which acts as a mathematical guarantee for non-divergence.
>
> These eventually lead to (i) peculiar training dynamics (as shown in Fig.3, whereas DoRA and LoRA show similar behavior), and (ii) better decoupling, supported by the strong robustness results in Fig.2 (previously Fig.4).
>
> In this regard, we notice that even if DeLoRA, by having a learnable boundary, in principle also has an unbounded Frobenius distance, in practice divergence does not happen, as shown in Fig.2 (previously Fig.4). This demonstrates that during finetuning, DeLoRA’s learnable boundary is able to effectively adjust and avoid divergence from the pretrained weights, behavior that does not happen with DoRA.”

---

### Author Response · Authors · 2024-12-02

We would like to thank all the reviewers for their detailed feedback and constructive inputs, which we believe led to extensive improvements, and substantially strengthened our final manuscript, as reflected by multiple increases in reviewer scores (pmk9,vsop,EKmU).

During the rebuttal phase we added to the main paper
- a new evaluation of our proposed DeLoRA method on the GLUE natural language understanding benchmark
- an extensive discussion on the core differences with DoRA, supported by DeLoRA’s peculiar training dynamics and better decoupling
- a new Euclidean distance plot to our robustness analysis, further showing DeLoRA’s non-divergent nature

In the Appendix, we further added
- a new ablation where we tested keeping DoRA’s magnitude term fixed, which interestingly didn’t lead to the same properties of DeLoRA
- a new robustness ablation on DeLoRA by separately varying its angular and strength components, which showed no loss of robustness
- new qualitative examples showing DeLoRA’s smaller degradation during prolonged finetuning

In addition, following the reviewers' feedback and concerns, we enhanced the manuscript's overall clarity. Part of these revisions include
- explicit mathematical formulations for our main ablation studies (in DeLoRA’s derivation from LoRA and ETHER)
- dedicated sections for DeLoRA's setup, the comparison with DoRA, and the robustness analysis
- additional remarks on DeLoRA’s initialization and stability

With the discussion period coming to an end, we believe that we have addressed all the comments from the reviewers which helped us improve our manuscript. We hope that includes the constructive and helpful comments of the reviewer nYwf whom we didn’t get to interact so far.

---

### Meta-Review · Area_Chair_6CtT · 2024-12-23

**Metareview:**

This paper proposes a variation on LoRA called DeLoRA which uses a learnable low-rank matrix with a unit norm, and a learnable magnitude. The authors also draw a connection between the proposed DeLoRA and multiplicative low-rank update schemes. On language and vision tasks, performance improvement from DeLoRA is reported.

Strengths: Clear, easy writing presenting a simple, practical approach to improving LoRA. Reviewer nTwf summarizes it well: "The ablation studies provide a convincing argument that somewhere in the continuum between LoRA and ETHER+ is a more powerful method."

Weaknesses: The idea of magnitude and direction decomposition for learning low-rank updates has already been proposed in DoRA - hence the novelty is not super clear. Well-established NLP and NLU tasks where LoRA has been frequently reported, such as the GLUE benchmarks and Commonsense Reasoning, is missing.

**Additional Comments On Reviewer Discussion:**

The main issues and changes made to the draft in response are very well summarized by the authors in the final thread. In particular, a new evaluation was added to align with existing LoRA literature; and an extensive discussion was added to separate the papers contributions from DoRA, a previous method. The only negatively inclined reviewer (vsop) acknowledged that their concerns were resolved and they were increasing their score (though this does not appear to be correctly reflected in the tool).

---

### Decision · Program_Chairs · 2025-01-22

Accept (Poster)